# Modification of Epoxy Coatings with Fluorocontaining Organosilicon Copolymers

**DOI:** 10.3390/polym16111571

**Published:** 2024-06-01

**Authors:** Dmitriy V. Krutskikh, Aleksey V. Shapagin, Irina O. Plyusnina, Nikita Yu. Budylin, Anna A. Shcherbina, Mikhail A. Soldatov

**Affiliations:** 1Department of Chemical Technology of Polymeric Composite Paints and Coatings, Mendeleev University of Chemical Technology, Miusskaya Sq. 9, 125047 Moscow, Russia; krutskikh.d.v@muctr.ru (D.V.K.); sherbina.a.a@muctr.ru (A.A.S.); 2Laboratory of Structural and Morphological Research, A.N. Frumkin Institute of Physical Chemistry and Electrochemistry of Russian Academy of Sciences, Leninskiy Pr. 31-4, 119071 Moscow, Russia; shapagin@mail.ru (A.V.S.); irinaplyusninar@yandex.ru (I.O.P.); budylin_nikita@mail.ru (N.Y.B.)

**Keywords:** epoxy, coatings, organosilicon, hydrophobicity, compatibility, fluorocontaining

## Abstract

Preparation of hydrophobic coatings is still a challenge for researchers in various fields of science. One of the easiest ways consists of the use of special modifiers. However, usually such modifiers are poorly compatible with organic polymeric matrixes, which leads to segregation of modifiers and deterioration of coating properties. In this work, we have synthesized a number of organosilicon copolymers and studied their compatibility with epoxy matrix and hydrophobic efficiency. It was shown that the increase of phenyl-containing units leads to increase of compatibility but decreases hydrophobic efficiency. Addition of small amounts of such modifiers into commercial epoxy paint material can lead to an increase of contact angle of the final coating from 63 to 87° without deterioration of other physico-mechanical properties. These results open new perspectives in preparation of organosilicon hydrophobic modifiers with directed properties for fields of application such as paints and coating materials.

## 1. Introduction

Preparation of stable hydrophobic coatings is of great interest in the field of modern materials science. Such coatings are widely used in various applications, such as aircraft manufacturing [1], shipbuilding [2], instrument making [3], anti-graffiti [4], and anti-icing coatings. Traditionally, hydrophobic coatings can be divided into three groups: hydrophobic (water contact angle θ > 90°); highly hydrophobic (θ > 120°); and superhydrophobic (θ > 150°) [5].

There are two general technological approaches to prepare a coating with low surface energy: surface modification, which consists of casting a modifier onto the surface of the final coating, and bulk modification [6], which consists of an addition of a modifier into the bulk of the initial material and further coating formation. The disadvantages of surface modification are the additional operation of casting of the hydrophobic agent to the surface of the product, probable uneven distribution of the hydrophobic agent over the surface of the coating, and the risk of low mechanical stability of the hydrophobic layer. The main disadvantage of bulk modification consists of low compatibility of the hydrophobic modifier with the polymeric matrix, which can lead to its phase segregation on the coating surface with deterioration of decorative and probably some physico-mechanical or physico-chemical properties of the coating [7,8].

To obtain hydrophobic coating, low-surface-energy materials such paraffins [9,10], fluorine-containing compounds [11,12,13], organosilicon compounds [14,15,16], and fluorinated organosilicon compounds [17,18,19,20,21] can be used. However, each of them has quite significant drawbacks. Coatings based on higher paraffins and fluorinated polymers have low adhesion to the substrate, which requires the additional use of a primer layer. Moreover, fluorinated materials have a high cost, which makes the production of additives and coatings only based on fluorinated polymers connections quite expensive [18,19]. The most promising materials for both surface and bulk modification are fluorinated organosilicon compounds, which show high hydrophobic properties and do not have the disadvantage of fluorinated compounds—low adhesion to the substrate. This advantage opens up wide possibilities for using these compounds. For example, in work [20], new UV-curable fluorine-containing organosilicon compounds were synthesized and characterized, and coatings based on these materials with a water contact angle of 121° and a water roll-off angle of 25° were also obtained. Works [21,22,23,24] describe in detail the preparation of a superhydrophobic coating based on fluorine-containing organosilicon compounds on a specially prepared aluminum substrate.

Nevertheless, the problem of compatibility of organosilicon modifiers with polymeric matrixes for the bulk modification remains relevant [25,26,27,28]. However, we are not aware of works dedicated to detailed study of the influence of molecular structure of fluorocontaining organosilicon copolymers on their compatibility with organic polymeric matrixes. Such a study might be promising for the development of novel stable hydrophobic coating materials. The aim of this work is to study phase equilibrium in binary systems based on fluorine-containing organosilicon compounds and epoxy polymeric matrix. To achieve this goal, we have synthesized a number of organosilicon copolymers having fluorinated groups, providing hydrophobicity, and phenyl groups, providing possible compatibility. It was found that the chemical structure influences both compatibility and hydrophobic efficiency of the synthesized copolymers.

## 2. Materials and Methods

### 2.1. Materials

The following materials were used in the work: bisphenol A-based epoxy resin ED-20 (epoxy, content of epoxy groups 20–22.5 wt.%) and and aliphatic amine hardener polyethylenepolyamine (PEPA) were purchased from Chimex LTD (Moscow, Russia), dimethyldimethoxysilane, methylphenyldimethoxysilane, acetic acid, dimethyldichlorosilane, methylphenyldichlorosilane, methyltrifluoropropyldichlorosilane were purchased from Thermo Scientific (Waltham, MA, USA), epoxy enamel EP-5287 was purchased from Kraskavo company (Moscow, Russia). Acetic acid was dried under P_2_O_5_ and distilled before the use.

### 2.2. Acidolytic Polycondensation of Dimethyldimethoxysilane and Methylphenyldimethoxysilane

For the synthesis of non-fluorinated hydrophobizing additives by acidolytic polycondensation, the dimethyldimethoxysilane, phenylmethyldimethoxysilane, and dry acetic acid were used. The amounts of materials are presented in Table 1.

The reaction scheme is shown in Figure 1.

Mixtures of silanes and acetic acid were stirred under refluxing in a round-bottom flask, equipped with a magnetic stirrer and reflux condenser with a calcium chloride tube, for 24 h. After that, the mixtures were dissolved in toluene and washed with distilled water until neutral medium (pH = 7) and dried with calcium sulfate. Finally, the solvent was evaporated with a vacuum rotary evaporator, and final products were dried under vacuum at 100 °C for 5 h.

### 2.3. Hydrolytic Polycondensation of Dimethyldichlorosilane and Methylphenyldichlorosilane

For the synthesis of non-fluorinated hydrophobizing additives by hydrolytic polycondensation, the dimethyldichlorosisilane, phenylmethyldichlorosisilane, toluene, and water were used. The amounts of materials are presented in Table 2.

The reaction scheme is shown in Figure 2.

The syntheses were carried out in a three-neck flask equipped with a magnetic stirrer, a reflux condenser with an alkaline tube, and a dropping funnel. In a typical synthetic procedure, 34 mL of water were added to the reaction flask. With intense stirring and cooling, a solution of 4.7 mL of dimethyldichlorosilane and 11.5 mL of methylphenyldichlorosilane in 34 mL of toluene were added dropwise, maintaining the temperature of the reaction at about 30–35 °C. After addition of chlorosilanes, the reaction mixture was stirred for another 15–20. The reaction mixture was then allowed to separate in two layers. The upper toluene layer containing the hydrolysate was separated, washed with a hot aqueous solution of sodium chloride (30 wt.% conc.) until neutral medium, and filtered, and the solvent was evaporated on a vacuum rotary evaporator at 60 °C until the hydrolysis products’ concentration reached about 60%. Next, the resulting solution was transferred to a round-bottomed flask with a reflux condenser and magnetic stirrer, and 40% aqueous solution of NaOH was added in an amount of 2 parts by weight per 100 parts by weight of the 60% solution of hydrolysis products. Polymerization was carried out at room temperature. The polymerization process was completed when a relative viscosity of 2.2–2.7 was achieved. To interrupt polymerization, the polymer was diluted with toluene and neutralized by adding concentrated hydrochloric acid dropwise until a weakly acidic reaction was attained. Excess hydrochloric acid was neutralized with calcium carbonate. The neutral solution was filtered, and excess solvent was evaporated.

### 2.4. Hydrolytic Polycondensation of Methyltrifluoropropidichlorosilane and Methylphenyldichlorosilane

For the synthesis of non-fluorinated hydrophobizing additives by hydrolytic polycondensation, the (3,3,3-Trifluoropropyl)methyldichlorosilane, phenylmethyldichlorosisilane, toluene, and water were used. The amounts of materials are presented in Table 3.

The reaction scheme is shown in Figure 3.

The synthesis procedure is similar to that described in the previous section.

### 2.5. Preparation of Compositions Based on Organosilicon Copolymers and Epoxy Resin

An epoxy resin solution was prepared in toluene with a concentration of 70, 50, and 30 wt.%. The aliphatic amine hardener PEPA was used with epoxy resin–hardener ratio of 1:1 by weight. A mixture of a solution of epoxy resin and a solution of the resulting copolymer in an appropriate solvent was applied to pre-cleaned glass plates. The coating was applied using a 100 µm-thick applicator. The coatings were dried in air for an hour and then at 100 °C for 1 h.

### 2.6. Characterizations

#### 2.6.1. Gel Permeation Chromatography

Analysis of molecular weight characteristics was carried out by gel-permeating chromatography on chromatograph “Shimadzu” (Kioto, Japan), sensor—RID refractometer—20 A, chromatographic column—SS SDV analytical 1000 A (300 × 8 mm), eluent—toluene.

#### 2.6.2. Nuclear Magnetic Resonance Spectroscopy

^1^H NMR spectra were obtained by nuclear magnetic resonance on a two-channel desktop nuclear magnetic resonance spectrometer NMReady-60PRO 1H/31P with a permanent magnet frequency of 60 MHz. CDCl_3_ was used as a deuterated solvent.

#### 2.6.3. Optical Interferometry

The study of solubility in binary systems was carried out by optical interferometry [28] on an ODA-2 diffusion interferometer (IPCE RAS, Moscow, Russia).

Epoxy was placed in a diffusion cell between glasses, the inner sides of which were coated with a translucent layer of Ni-Cr alloy. An angle of ≤2° was set between the glasses, which was necessary for the appearance of an interference picture when passing a laser beam with λ = 635 nm. Then, the space between the glasses was filled with copolymer.

In order to obtain information about phase equilibria in systems, studies were carried out in the mode of stepwise heating and cooling with a step of 10 °C in the temperature range from 130 to 20 °C. At each stage, the system was thermostated to a state of equilibrium, and interferograms were recorded. The moment of contact of the components was considered the beginning of the interdiffusion process. The interference picture in the interdiffusion zone was characterized by the curvature of isoconcentration bands in accordance with the change in the refractive index in the mixture of the studied components. By processing the mutual diffusion zones of the interferograms, concentration profiles were obtained. A smooth change in concentration in the diffusion zone from one component to another indicated the homogeneity of the system at a given temperature. Concentration jumps in the profiles determined the compositions of coexisting phases corresponding to the points of the binodal curve of the phase diagram [29,30,31].

#### 2.6.4. Refractometry

To construct a phase diagram from interference pictures, information about the total number of interference lines (N) is required. To calculate N using equation 1, the temperature dependencies of the refractive index of the initial components (n_1_, n_2_) are required. The increment of the refractive index (△ = 0.003) per line is a constant and corresponds to the angle between the glasses equal to 2°.
N = (n_1_ − n_2_)/△(1)

The concentration per line is calculated using the next equation:C = 1/N(2)

The studies were carried out on an Abbe ATAGO NAR-2T refractometer (“Atago Co. Ltd.”, Tokyo, Japan) in the “heating-cooling” mode in the temperature range from 20 to 60 °C with an accuracy of ±0.0001.

#### 2.6.5. Sessile Drop Method

The contact angle values were obtained by the sessile drop method using an FM 40 Mk 2 EasyDrop device (Kruss, Hamburg, Germany); the contact angle was measured after 30 s presence of a drop on the surface of the coating.

## 3. Results and Discussion

In the first part of the work, homo- and copolymers with different ratios of dimethylsiloxane and methylphenylsiloxane were obtained, which were characterized by NMR and GPC. Then, their compatibility with the epoxy matrix was studied using optical interferometry, and the water contact angles on the resulting coatings were measured. NMR spectra of the obtained compounds are presented in Figure 4.

The spectra of SiO2–5 samples contain signals from both methylsilyl groups and phenylsilyl groups. In the spectrum of the SiO1 sample there are no signals of phenyl groups, and in the spectrum of SiO6 there are no signals of dimethylsiloxane groups. The ratios of integral intensities were close to theoretical ones, which indicates the successful synthesis of the modifiers. The NMR spectra of organosilicon polymers obtained by hydrolytic polycondensation (Figure 5) are similar to the NMR spectra of polymers obtained by acidolytic polycondensation. They also clearly show the characteristic peaks of the phenyl substituent, the signal intensity of which increases with increasing their content in the polymer.

The resulting NMR spectra of fluorinated copolymers (Figure 6) contain a larger number of peaks due to a change in one of the substituents: the methyl substituent was replaced by trifluoropropyl. Therefore, additional peaks appeared in the 1H NMR spectra, indicating the presence of different CH_2_ groups: in the region of 1 ppm CH_2_ with silicon and in the region of 2 ppm CH_2_ between two carbohydrates. The remaining peaks are similar to those obtained in the previous spectra.

The study of the mutual solubility of fluorinated copolymers and epoxy oligomer was carried out using optical interferometry. To determine the compositions of coexisting phases using this method, the temperature dependencies of the refractive index of the individual components of the mixture are necessary (Figure 7). Since all measurements were carried out within the same phase state, the dependencies have a linear form.

Analysis of the diffusion zones of interference pictures for the epoxy–SiO1 system (Figure 8) showed that when semi-infinite media are combined, the system is characterized by a phase boundary. Note that to the left and to the right of the phase boundary, over the entire temperature range, no bending of the interference lines associated with a change in the refractive index in the diffusion zone is observed. Reducing the temperature to room temperature leads to the formation of phase particles on both sides of the phase boundary. This indicates partial solubility of the components at extremely low concentrations. Thus, it can be stated that the components of the epoxy–SiO1 system are practically insoluble in each other.

Interdiffusion zones of the epoxy–SiO2 system (Figure 9) are characterized by a gradient of the refractive index (concentration) on both sides of the phase boundary at which a concentration jump occurs. This is identified at elevated temperatures, indicating comparatively better compatibility of the components compared to the epoxy–SiO1 system. A decrease in temperature leads to the appearance of dispersed particles enriched in SiO2 to the left of the phase boundary and enriched in epoxy to the right. As the temperature decreases, the heterogeneous zone expands, which classifies the systems into the class of amorphous stratification systems characterized by the upper critical solution temperature (UCST).

In contrast to the mixtures of epoxy with SiO1 and SiO2 discussed above, in which the UCST was significantly higher than the maximum mixing temperature of 120 °C, for the epoxy–SiO3 system it was possible to determine the critical temperature equal to 46 °C (Figure 10). In Figure 11, the information obtained from the analysis of interference pictures is generalized on the temperature-concentration field in the form of binodal curves of phase diagrams. It has been established that the introduction of phenyl fragments into the reaction mixture when preparing an organosilicon copolymer improves solubility in the epoxy–SiO system. Mixing epoxy with SiO4, SiO5, and SiO6 showed complete solubility of the components of the mixtures over the entire studied temperature range from 20 to 120 °C.

Studies of diffusion zones in binary gradient solutions of the epoxy–FSiO system were carried out in the temperature range from 60 to 120 °C. Temperature restrictions are due to the fact that above 120 °C the copolymer is destroyed, and below 60 °C the epoxy oligomer is in a glassy state. It was established (Figure 12) that a decrease in the concentration of (3,3,3-trifluoropropyl)methyldichlorosilane and an increase in the concentration of methylphenyldichlorosilane in the reaction mixture when preparing the copolymer was accompanied by an improvement in the mutual solubility of the components, which is shown by the relative position of the binodal curves of the epoxy–FSiO3 and epoxy–FSiO4 systems on the temperature-concentration field of the phase diagram (Figure 13). Note that the combination of epoxy with fluorinated organosilicon components, as in the case of combination with non-fluorinated ones (discussed above), is described by amorphous separation diagrams with UCST.

It is important that the interference pattern shown in Figure 14 characterizes the components of the epoxy–FSiO5 system as completely soluble in each other over the entire studied temperature range.

Figure 15 shows typical kinetic dependencies characterizing the movement of isoconcentration planes in the epoxy–FSiO5 system. It can be seen that the dependencies in diffusion coordinates X ≈ k√t, where k is a constant characterizing the partial diffusion coefficients of the components, are described by a linear dependence. A similar type of movement of isoconcentration planes was observed over the entire selected temperature range. This suggests that the mechanism of mixing components in the epoxy–FSIO system is diffusion.

Note that the concentration profile is asymmetrical (Figure 16a), and the coefficients of interdiffusion of components (Figure 16b) calculated from it lie in the range from 6.3 × 10^−7^ (in the region of diluted epoxy solutions) to 6.3 × 10^−8^ cm^2^/s (in the region of epoxy-concentrated solutions).

After studying the compatibility of the obtained organosilicon copolymers with the epoxy matrix, it was decided to study the contact angles only on coatings modified with copolymers of compositions 4 and 5 due to their better compatibility. Contact angle measurements for polymer 6 were not carried out due to the fact that it contains only methylphenylsiloxane units and has the least hydrophobizing property of all the synthesized copolymers. We have 2 wt.% concentration of modifiers because at higher concentration, the hydrophobicity almost does not change, and at the lower concentrations the contact angle values are lower as well.

The results of measuring the contact angles of water wetting on the resulting coatings are presented in Figure 17, Figure 18, Figure 19 and Figure 20.

For modifiers obtained by hydrolytic polycondensation, the dependence described above also holds. The values of contact angles are quite close to those obtained by modification with compounds obtained by acidolytic polycondensation.

Based on previous experiments on combining fluorine-containing organosilicon polymers with an epoxy matrix, it was decided to further use water-repellent agents FSiO4 and FSiO5 due to their better compatibility and high water-repellent properties.

As can be seen from the obtained data, modification of the epoxy resin leads to an increase in the contact angle. With a decrease in the content of dimethylsiloxane fragments, the hydrophobizing ability of the modifier decreases. Such results correlate with previous studies, where it was shown that fluorinated organosilicon copolymers can be partially segregated on the surface of coatings with the formation of hydrophobic layers [18,19].

So, it has been shown that fluorinated organosilicon copolymers have better hydrophobizing ability than non-fluorinated copolymers.

For further investigations and study of application aspects of the synthesized copolymers, we have used commercial epoxy paint EP-5287 (Russia) of grey color. It was modified with fluorine-containing organosilicon polymers FSiO3 and FSiO4 in an amount of 2 wt.%. After that, the physical and mechanical properties of the resulting coatings and their contact angles were studied.

The test results are presented in Table 4.

The results of the paint adhesion test are presented in Figure 21.

Based on the data obtained, we can conclude that the addition of a water repellent does not affect the physical and mechanical properties of the coating.

The water contact angles of the coating deposited on a metal substrate were also measured. The results are presented in Figure 22 and Figure 23.

Additionally, we have measured surface energies for our coatings. Surface energies were calculated by the Owens–Wendt method with the use of water and ethylene glycol as testing liquids (Table 5). According to the results, the addition of the modifiers leads to a decrease of the surface free energy γ as well as its dispersive γ^d^ and polar γ^p^ parts. Moreover, surface energy decreases with the increase of content of fluorinated units in the modifier.

So, it has been shown that the addition of the obtained fluorinated organosilicon polymers to epoxy enamel makes it possible to increase the water-repellent properties of the coating without changing its physical and mechanical properties.

## 4. Conclusions

A number of organosilicon copolymers containing dimethylsiloxane, methyltrifluoropropylsiloxane, and methylphenylsiloxane units have been synthesized. It has been shown that the increase of methylphenylsiloxane units content increases the compatibility of the synthesized organosilicon copolymers with the epoxy matrix. It was also shown that the introduction of 2 wt.% of synthesized copolymers can increase the contact angle of coatings by 20° in the case of using copolymers containing dimethylsiloxane and methylphenylsiloxane units and by 30° in the case of using copolymers containing methyltrifluoropropylsiloxane and methylphenylsiloxane units. The most optimal in terms of compatibility and hydrophobizing properties are copolymers with a ratio of dimethylsiloxane/methyltrifluoropropylsiloxane units to methylphenylsiloxane units equal to 4:6 2:8. Using the example of commercial epoxy paint EP-5287, it was shown that the synthesized modifiers lead to an increase of the contact angle from 63° to 87° without any deterioration of decorative and physico-mechanical properties such as: adhesion, hardness, impact strength, elasticity, and drying time of the coating. It was also shown that the increase of content of fluorinated units in modifiers decreases the surface energy of modified coatings. The contact angle does not exceed 90° and the obtained coatings technically are not hydrophobic, but the obtained values of contact angle are enough for some practical application. It is in plans for future research to study the influence of molecular weight, molecular topology of modifiers, and other factors on hydrophobic efficiency, which probably might increase the contact angle values.

## Figures and Tables

**Figure 1 polymers-16-01571-f001:**
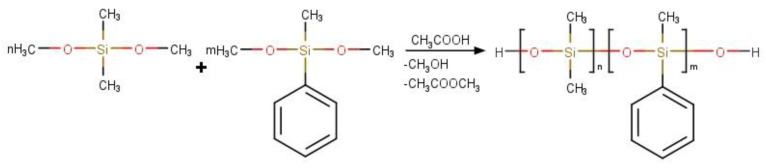
Reaction scheme of acidolytic polycondensation of organoalkoxysilanes.

**Figure 2 polymers-16-01571-f002:**
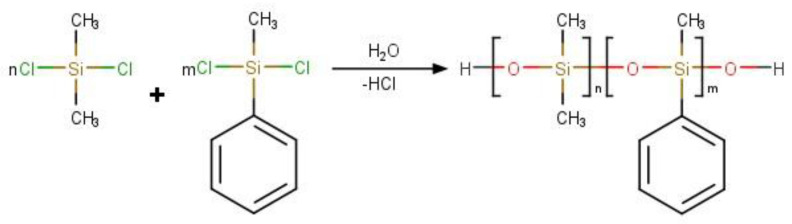
Reaction scheme of hydrolytic polycondensation of organochlorosilanes.

**Figure 3 polymers-16-01571-f003:**
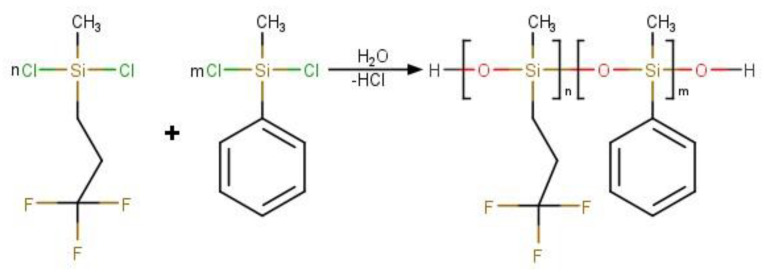
Scheme of the reaction of hydrolytic polycondensation of organochlorosilanes with formation of fluoro-containing copolymers.

**Figure 4 polymers-16-01571-f004:**
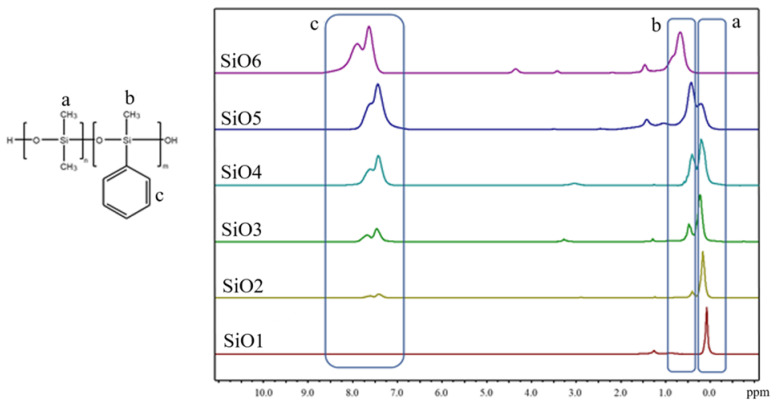
NMR spectra of the obtained compounds.

**Figure 5 polymers-16-01571-f005:**
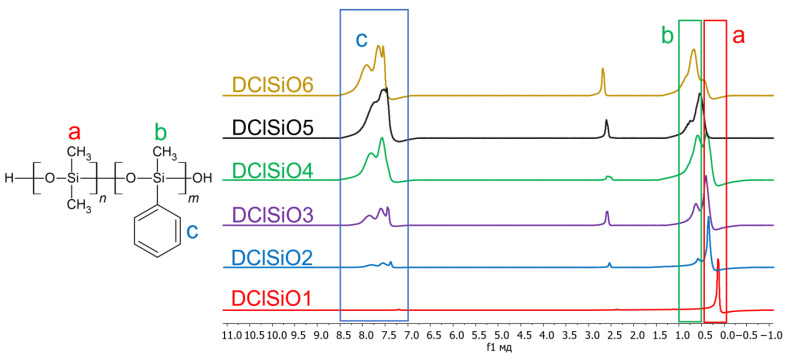
NMR spectra of the obtained compounds.

**Figure 6 polymers-16-01571-f006:**
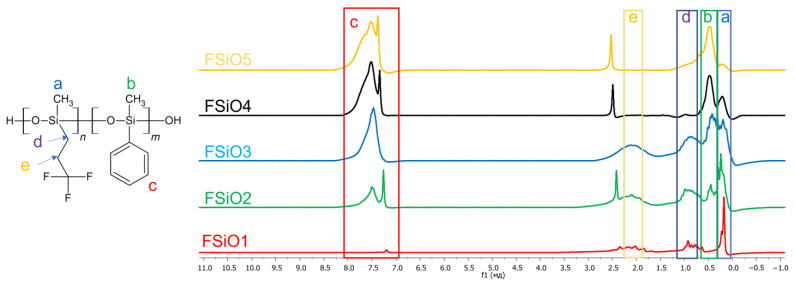
NMR spectra of the obtained compounds.

**Figure 7 polymers-16-01571-f007:**
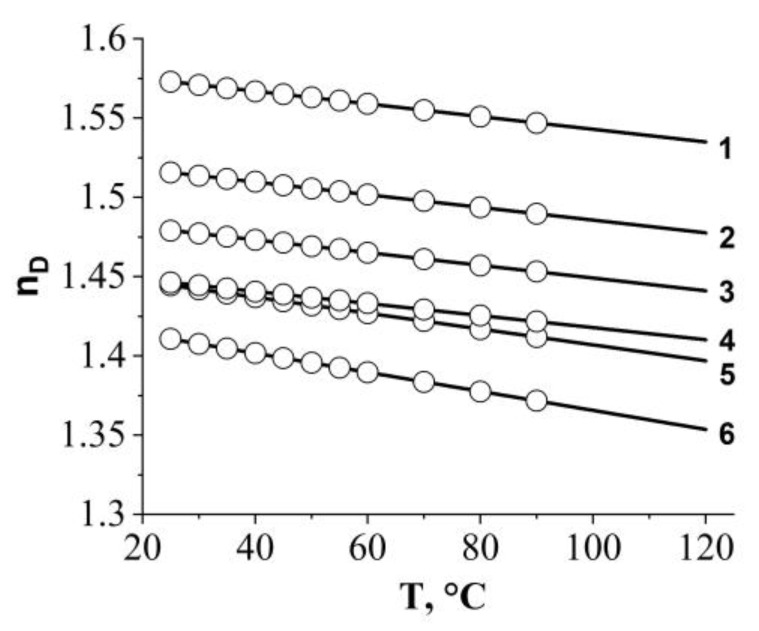
Temperature dependence of the refractive index for: 1—epoxy, 2—FSiO5, 3—FSiO4, 4—FSiO3, 5—SiO2, 6—SiO1.

**Figure 8 polymers-16-01571-f008:**
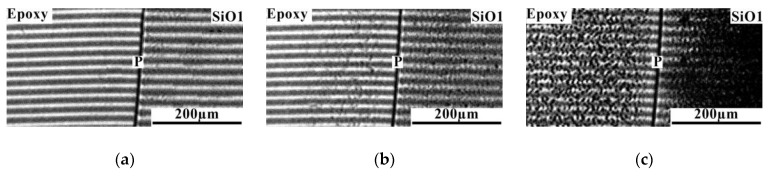
Interferograms of the interdiffusion zone of the epoxy–SiO1 system at temperature: (**a**) 130; (**b**) 110; (**c**) 20 °C. **P** is the phase boundary.

**Figure 9 polymers-16-01571-f009:**
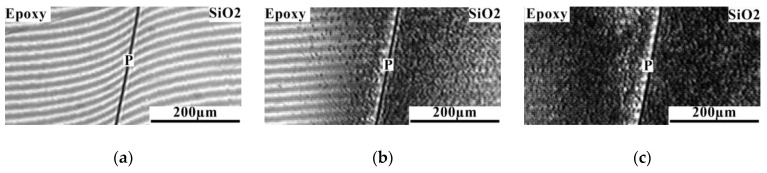
Interferograms of the interdiffusion zone of the epoxy–SiO2 system at temperature: (**a**) 130; (**b**) 64; (**c**) 20 °C. **P** is the phase boundary.

**Figure 10 polymers-16-01571-f010:**
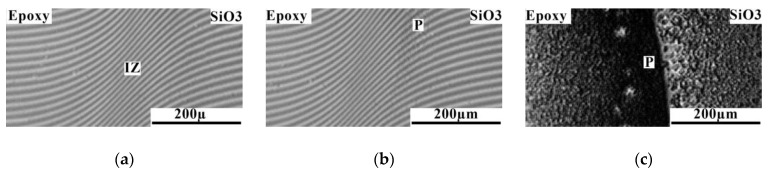
Interferograms of the interdiffusion zone of the epoxy–SiO3 system at temperature: (**a**) 50; (**b**) 46; (**c**) 20 °C. **IZ** is the interdiffusion zone, **P** is the phase boundary.

**Figure 11 polymers-16-01571-f011:**
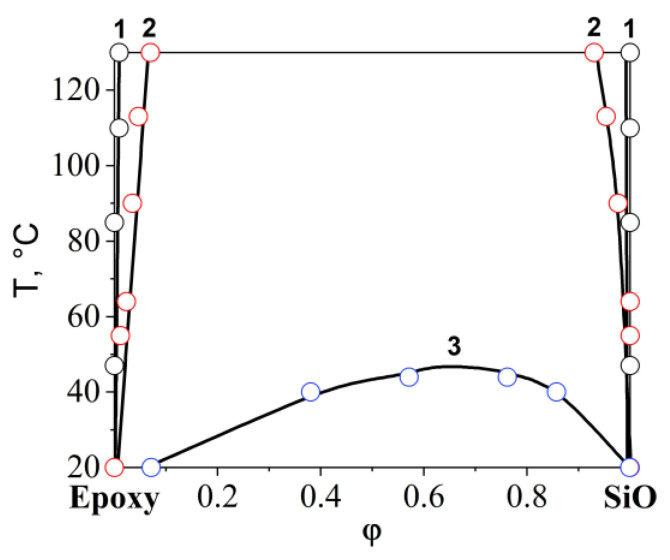
Phase diagram of systems: 1—epoxy–SiO1, 2—epoxy–SiO2, 3—epoxy–SiO3.

**Figure 12 polymers-16-01571-f012:**
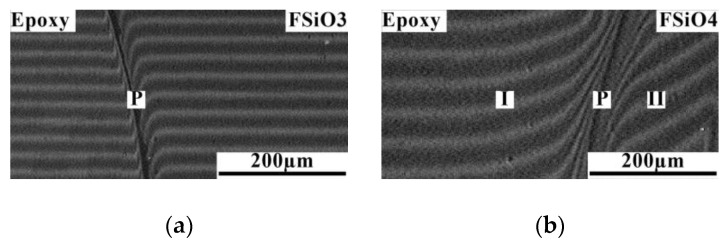
Interferograms of the interdiffusion zone of the system: (**a**) epoxy–FSiO3; (**b**) epoxy–FSiO4 at T = 60 °C. I—diffusion zone of epoxy in FSiO3, II—diffusion zone of FSiO3 in epoxy, **P** is the phase boundary.

**Figure 13 polymers-16-01571-f013:**
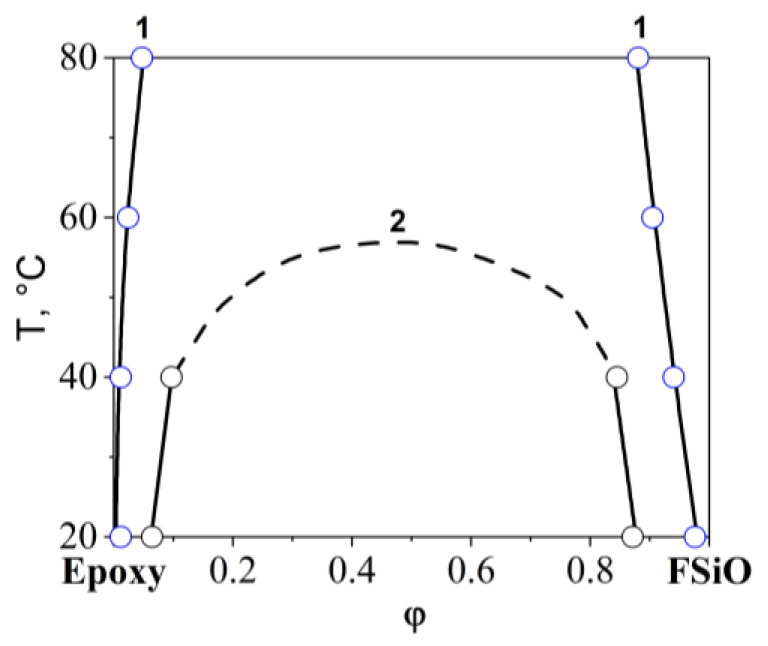
Phase diagram of systems: 1—epoxy–FSiO3, 2—epoxy–FSiO4.

**Figure 14 polymers-16-01571-f014:**
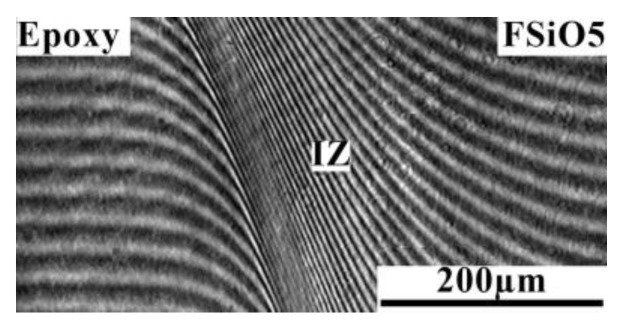
Interferogram of the interdiffusion zone of the epoxy–FSiO5 system at a temperature of 20 °C and a time of 8 min. **IZ** is the interdiffusion zone.

**Figure 15 polymers-16-01571-f015:**
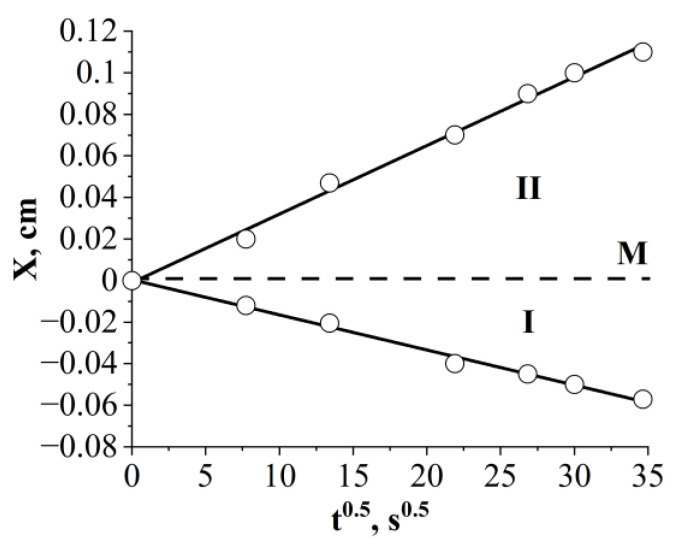
Kinetic dependences of the movement of isoconcentration planes of the epoxy–FSiO5 system at a temperature of 20 °C. **I** corresponds to the diffusion front of FSiO5 in epoxy, **II** corresponds to the diffusion front of epoxy in FSiO5, **M** is the Matano–Boltzmann plane.

**Figure 16 polymers-16-01571-f016:**
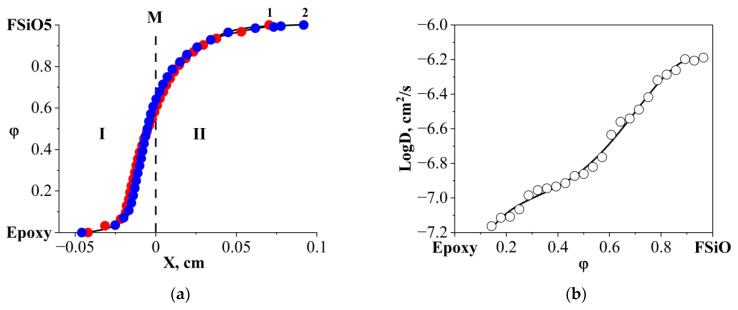
Concentration dependences of the epoxy–FSiO5 system at a temperature of 20 °C: (**a**) interdiffusion profile at 1–8, 2–12 min. **I** corresponds to the diffusion front of FSiO5 in epoxy, **II** corresponds to the diffusion front of epoxy in FSiO5, **M** is the Matano–Boltzmann plane; (**b**) interdiffusion coefficients.

**Figure 17 polymers-16-01571-f017:**
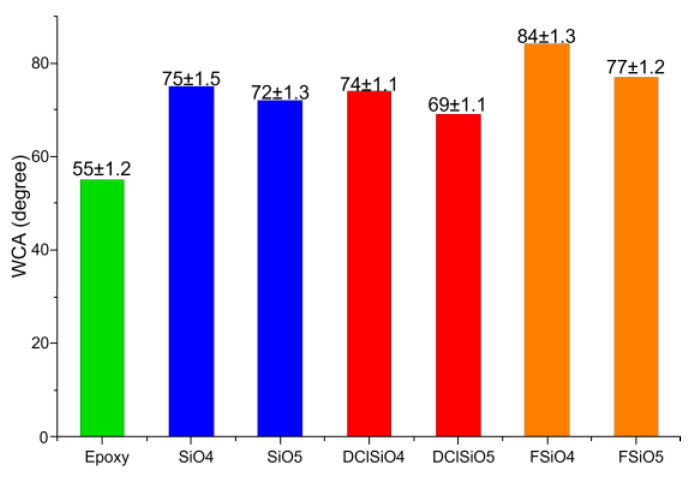
Contact angles of epoxy coatings, modified with 2 wt.% of organosilicon copolymers.

**Figure 18 polymers-16-01571-f018:**
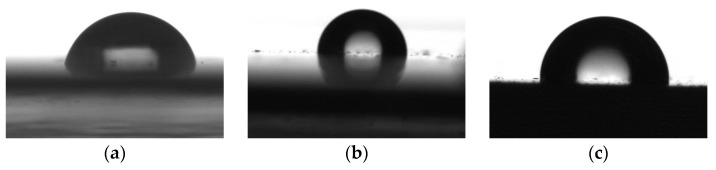
Images of water droplets on the surface of coatings, modified with SiO copolymers; (**a**) epoxy; (**b**) epoxy + SiO4; (**c**) epoxy + SiO5.

**Figure 19 polymers-16-01571-f019:**
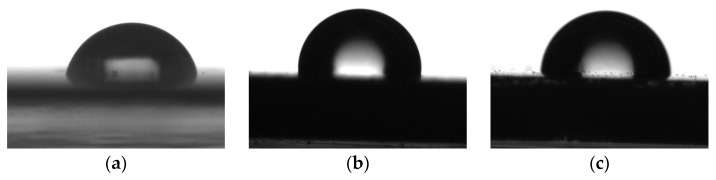
Images of water droplets on the surface of coatings, modified with DClSiO copolymers; (**a**) epoxy; (**b**) epoxy + DClSiO4; (**c**) epoxy + DClSiO5.

**Figure 20 polymers-16-01571-f020:**
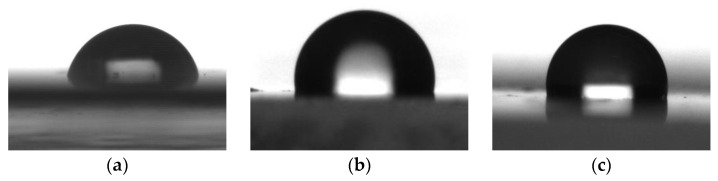
Images of water droplets on the surface of coatings, modified with FSiO copolymers; (**a**) epoxy; (**b**) epoxy + FSiO4; (**c**) epoxy + FSiO5.

**Figure 21 polymers-16-01571-f021:**
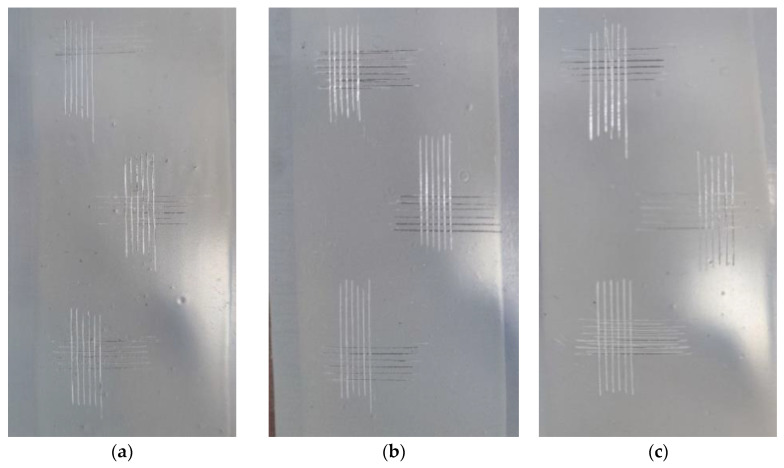
Results of paintwork adhesion testing, (**a**) EP-5287; (**b**) EP-5287 + FSiO3; (**c**) EP-5287 + FSiO4.

**Figure 22 polymers-16-01571-f022:**
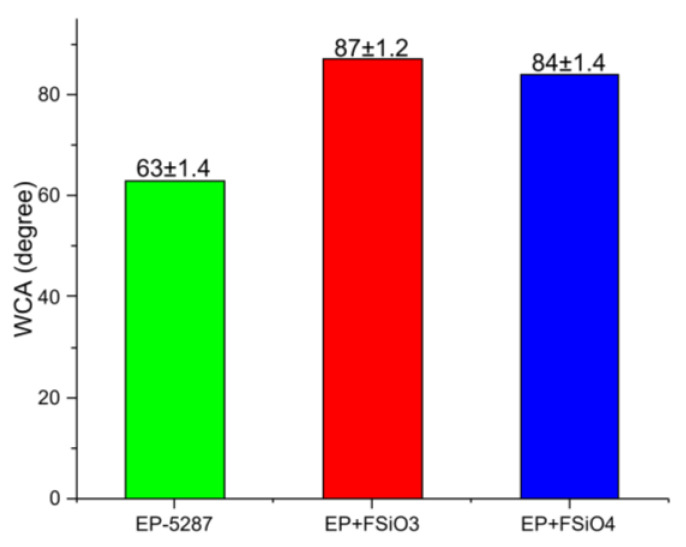
Contact angles of EP-5287-modified coatings.

**Figure 23 polymers-16-01571-f023:**
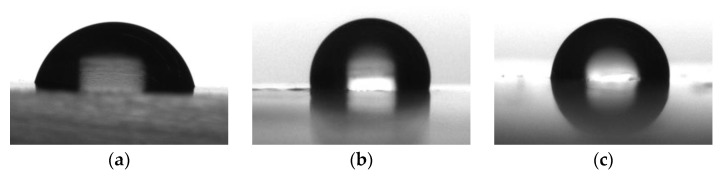
Images of water droplets on the surface of EP-5287 coatings, modified with FSiO copolymers: (**a**) EP-5287; (**b**) EP-5287 + FSiO3; (**c**) EP-5287 + FSiO4.

**Table 1 polymers-16-01571-t001:** Compositions of reaction mixtures for the synthesis of organosilicon copolymers by acidolytic polycondensation.

Sample	Molar Ratios of Reagents		
	Dimethyldiethoxysilane	Methylphenyldimethoxysilane	Acetic Acid
SiO1	10	0	30
SiO2	8	2	30
SiO3	6	4	30
SiO4	4	6	30
SiO5	2	8	30
SiO6	0	10	30

**Table 2 polymers-16-01571-t002:** Compositions of reaction mixtures for the synthesis of organosilicon copolymers by hydrolytic polycondensation.

Sample	Molar Ratios of Reagents	
	Dimethyldichlorosilane	Methylphenyldichlorosilane
DClSiO1	10	0
DClSiO2	8	2
DClSiO3	6	4
DClSiO4	4	6
DClSiO5	2	8
DClSiO6	0	10

**Table 3 polymers-16-01571-t003:** Compositions of reaction mixtures for the synthesis of fluorinated organosilicon copolymers by hydrolytic polycondensation.

Sample	Molar Ratios of Reagents	
	(3,3,3-Trifluoropropyl)methyldichlorosilane	Methylphenyldichlorosilane
FSiO1	10	0
FSiO2	8	2
FSiO3	6	4
FSiO4	4	6
FSiO5	2	8
FSiO6	0	10

**Table 4 polymers-16-01571-t004:** Physical and mechanical properties of coatings.

Parameter	Testing Method	EP-5287	EP-5287 + FSiO3	EP-5287 + FSiO4
**Film adhesion by cross-cut test, score**	ISO 2409:2013	0	0	0
**Relative hardness according to a pendulum device (König pendulum)**	ISO 1522:2022	0.5	0.5	0.5
**Drying time of the film, at the temperature of 60 °C, h**	ISO 9117-3:2010	1	1	1
**Film strength upon impact, cm**	ISO 6272-1:2002	65	65	65
**Film elasticity by bend test, mm**	ISO 1519:2002	2	2	2

**Table 5 polymers-16-01571-t005:** Surface energies of modified coatings.

Sample Name	γ^d^, mJ/m^2^	γ^p^, mJ/m^2^	γ, mJ/m^2^	R-Square
**Epoxy**	9.60	35.44	45.04	0.9997
**Epoxy + FSiO4**	18.31	7.19	25.50	0.9058
**Epoxy + FSiO5**	19.60	11.18	30.78	0.9946
**EP-5287**	12.67	25.64	38.31	0.9997
**EP-5287 + FSiO3**	20.06	5.38	25.44	0.9389
**EP-5287 + FSiO4**	20.55	6.81	27.36	0.9975

## Data Availability

Data are contained within the article.

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
