# Peer review of "Modification of Epoxy Coatings with Fluorocontaining Organosilicon Copolymers"

_polymers, 2024, doi:10.3390/polym16111571_

Round 1

Reviewer 1 Report

Comments and Suggestions for Authors

Authors studied the Modification of epoxy coatings with fluorocontaining organosilicon copolymers. The research findings are interesting. however, some of the points need to be addressed to improve the quality of the paper

1. Include the novelty statement at the end of the introduction section

2. Fig 18 to 23, magnification/scale bar missing in the images. please include it

3. In Fig. 18-20 Mode discussion with technical reasons behind it is missing. Discuss in details with some references.

4. Addition of the obtained fluorinated organosilicon 346 polymers to epoxy enamel makes it possible to increase the water-repellent properties of 347 the coating without changing its physical and mechanical properties. please justify with some references.

Author Response

1. Include the novelty statement at the end of the introduction section

Thank you for this notice. We have added this text into introduction: However, we are not aware about works, dedicated to detailed study of influence of molecular structure of fluorocontaining organosilicon copolymers on their compatibility with organic polymeric matrixes. Such study might be promising for development of novel stable hydrophobic coating materials. 

2. Fig 18 to 23, magnification/scale bar missing in the images. please include it

Thank you for this notice. Unfortunately the software, we used for contact angle measurements, does not allow to add the scale bar. And images of the coatings were also made on the phone camera without possibility to add scale bar. We hope that reviewer will let us to use these figures without changings.

3. In Fig. 18-20 Mode discussion with technical reasons behind it is missing. Discuss in details with some references.

Thank you for your notice. Such results correlate with previous studies, where it was shown, that fluorinated organosilicon copolymers can be partially segregated on the surface of coatings with formation of hydrophobic layers[16,17].

4. Addition of the obtained fluorinated organosilicon 346 polymers to epoxy enamel makes it possible to increase the water-repellent properties of 347 the coating without changing its physical and mechanical properties. please justify with some references.

Sorry, but we do not quite understand what exactly we should justify, cause it is original results of our work showing that our hydrophobic agent do not deteriorate mechanical properties. We hope that reviewer let us do not add references

Reviewer 2 Report

Comments and Suggestions for Authors

The manuscript “Modification of epoxy coatings with fluorocontaining organo-silicon copolymers” is a well-written research work. It describes the development of a novel epoxy-based hydrophobic coating by implementing organosilicon copolymers. However, there are a few minor revisions necessary.

Line 26: Is anti-vandal a proper term? Please clarify in a few words, e.g. in brackets

Line 39-40: Specify what is meant by Probably? No source referenced. Please expand or add a reference to back up the claim.

Line 48: No space after the dot. Please check out through-out the text for some minor inaccuracies.

Line 52: Extra space after [20]. Check the rest of the manuscript for similar inaccuracies.

Line 67: epoxy resin (Epoxy) – which epoxy was used?

Figure 1, 2, and 3: Please consider including a higher resolution Figure/Scheme.

Line 103: Please use . instead of , for decimals.

Line 108: 30% conc. – molar/by weight? Specify.

Line 132: Similar, no difference?

// Minor Revision.

Author Response

Line 26: Is anti-vandal a proper term? Please clarify in a few words, e.g. in brackets

Thank you for your question. Sometimes the term "anti-vandal" is used, but here we have replaced it with "anti-graffity" term and have added reference.

Line 39-40: Specify what is meant by Probably? No source referenced. Please expand or add a reference to back up the claim.

Thank you for your notice. We have added references [7,8] where it was shown that phase segregation may cause deterioration of psysico-mechanical properties of polymer composites.

Line 48: No space after the dot. Please check out through-out the text for some minor inaccuracies.

Thank you for your notice. We have carefully checked the text on such misprints.

Line 52: Extra space after [20]. Check the rest of the manuscript for similar inaccuracies.

Thank you for your notice. We have carefully checked the text on such misprints.

Line 67: epoxy resin (Epoxy) – which epoxy was used?

Thank you for your question. We used bisphenol A based epoxy resin ED-20 (Russia, content of epoxy groups 20-22.5 wt. %) 

Figure 1, 2, and 3: Please consider including a higher resolution Figure/Scheme.

Thank you for your notice. We have replaced Figures with better ones.

Line 103: Please use . instead of , for decimals.

Thank you for your notice. We have carefully checked the text on such mistakes.

Line 108: 30% conc. – molar/by weight? Specify.

Thank you for your question. Here we used weight concentration.

Line 132: Similar, no difference?

Thank you for your question. Yes, the methodology of synthesis was the same with difference only in the used monomers.

Reviewer 3 Report

Comments and Suggestions for Authors

The manuscript describes;  hydrophobizing organosilicon copolymers with various ratios of dimethylsiloxane and methylphenylsiloxane are synthesized and characterized. Optical interferometry studies revealed that increasing the phenyl content in the organosilicon copolymers improved compatibility with the epoxy matrix. The results indicated different levels of compatibility between epoxy resin and the synthesized copolymers. The Epoxy-SiO1 system displayed limited solubility, while the Epoxy-SiO2 system showed improved compatibility at higher temperatures. Further results indicated complete solubility for Epoxy-SiO4, SiO5, and SiO6 systems, suggesting that higher phenyl content improves compatibility.

Studies of contact angles demonstrated that modifying epoxy resin with organosilicon copolymers leads to an increase in hydrophobicity. Fluorinated copolymers showed better hydrophobic properties than non-fluorinated copolymers, with coatings based on fluorinated compounds achieving higher contact angles. Physical and mechanical properties of epoxy coatings modified with fluorinated copolymers showed no significant changes in film adhesion, hardness, drying time, impact strength, or elasticity.

There are some Queries to be answered:

1.        Does the addition of organosilicon copolymers result in reduced adhesion of the coating to the substrate?

2.        Do fluorinated organosilicon copolymers increase the drying time of epoxy coatings?

3.        Does the modification of epoxy coatings with fluorinated copolymers lead to a significant loss of elasticity?

4.        Can organosilicon copolymers cause phase segregation in epoxy-based coatings?

5.        Is there a noticeable reduction in mechanical strength when epoxy coatings are modified with fluorinated organosilicon copolymers?

     In this study on the effect of organosilicon copolymers on epoxy-based coatings, several observations were explored:

·        The addition of organosilicon copolymers did not result in reduced adhesion of the coating to the substrate.

·        Cross-cut tests showed consistent results, indicating that the adhesion remained intact. Modifying epoxy coatings with fluorinated organosilicon copolymers did not increase the drying time.

·        The ISO 9117-3:2010 test showed similar drying times across samples, suggesting that the modification had no significant impact. No significant loss of elasticity was observed when epoxy coatings were modified with fluorinated copolymers.

·        Bend tests demonstrated similar elasticity values, indicating that the coatings maintained their flexibility.

·        Although phase segregation is a common concern with the addition of organosilicon modifiers, the study found that the chosen copolymers were compatible with the epoxy matrix, avoiding phase segregation. The impact and hardness tests revealed that the mechanical strength of the epoxy coatings did not deteriorate upon the addition of fluorinated copolymers. This suggests that the physical and mechanical properties remained stable despite the modification.

Comments on the Quality of English Language

Moderately be improved

Author Response

  1. Does the addition of organosilicon copolymers result in reduced adhesion of the coating to the substrate?
    Thank you for your question. According to results adhesion remains the same after modification.
  2. Do fluorinated organosilicon copolymers increase the drying time of epoxy coatings?
    Thank you for your question. The modifier according the results does not influence on drying time
  3. Does the modification of epoxy coatings with fluorinated copolymers lead to a significant loss of elasticity?
    Thank you for your question. According to results elasticity remains the same after modification.
  4. Can organosilicon copolymers cause phase segregation in epoxy-based coatings?
    Thank you for your question. Phase segregation can be caused by various factors, such as chemical structure, molecular weight of modifier and cross-linking density of obtained coatings. Some of these factors are in plans to be studied in future researches. 
  5. Is there a noticeable reduction in mechanical strength when epoxy coatings are modified with fluorinated organosilicon copolymers?
    Thank you for your question. According to results the impact strength remains the same after modification.

Reviewer 4 Report

Comments and Suggestions for Authors

The article is well written. However, there are several suggestion to improve the paper.

1. Please add more detail on contact angle study. For example, surface energy and add the standard deviation (error bar) for each sample. You may make 3 or more observations on each surface with different location.

2. What is the rationale of using 2 wt%?

3. Conclusion could be more thorough. Please elaborate more than the impact on contact angle.

4. The best effort only exhibits 87 degree contact angle. Hence, it is not even hydrophobic. Could please elaborate more on this matter?

5. The abstract is shown different result from the rest of the paper. In abstract, the improvement is 63° to 90° while the rest is 63° to 87°.

Author Response

1. Please add more detail on contact angle study. For example, surface energy and add the standard deviation (error bar) for each sample. You may make 3 or more observations on each surface with different location.

Thank you for your suggestion. We have measured surface energies for our coatings and added error bars.

2. What is the rationale of using 2 wt%?

Thank you for this question. Preliminary we have made several concentrations and have found that 2 wt.% was optimal, because at lower concentrations the contact angles were loer as well and higher concentrations did not give significant increase in contact angle.

3. Conclusion could be more thorough. Please elaborate more than the impact on contact angle.

Thank you for your notice. We have modified the conclusions.

4. The best effort only exhibits 87 degree contact angle. Hence, it is not even hydrophobic. Could please elaborate more on this matter?

Thank you for your notice. Despite the contact angle does not exceed 90° and the obtained coatings technically are not hydrophobic, but the obtained values of contact angle are enough for some practical application. It is in plans for future researches to study influence of molecular weight, molecular topology of modifiers and and other factors on hydrophobic efficiency which probably might increase the contact angle values.

5. The abstract is shown different result from the rest of the paper. In abstract, the improvement is 63° to 90° while the rest is 63° to 87°.

Thank you for your notice. Indee, we have made mistake in the abstract. We have corrected it.

Round 2

Reviewer 3 Report

Comments and Suggestions for Authors

In a study on the effect of organosilicon copolymers on epoxy-based coatings, several negative aspects were explored:

The addition of organosilicon copolymers did not result in reduced adhesion of the coating to the substrate. Cross-cut tests showed consistent results, indicating that the adhesion remained intact. Modifying epoxy coatings with fluorinated organosilicon copolymers did not increase the drying time. The ISO 9117-3:2010 test showed similar drying times across samples, suggesting that the modification had no significant impact. No significant loss of elasticity was observed when epoxy coatings were modified with fluorinated copolymers. Bend tests demonstrated similar elasticity values, indicating that the coatings maintained their flexibility.

Although phase segregation is a common concern with the addition of organosilicon modifiers, the study found that the chosen copolymers were compatible with the epoxy matrix, avoiding phase segregation. The impact and hardness tests revealed that the mechanical strength of the epoxy coatings did not deteriorate upon the addition of fluorinated copolymers. This suggests that the physical and mechanical properties remained stable despite the modification.

Overall, the results suggest that while certain organosilicon copolymers can improve the hydrophobicity of epoxy coatings, they do not cause adverse effects such as reduced adhesion, increased drying time, and loss of elasticity, phase segregation, or decreased mechanical strength.

Comments on the Quality of English Language

Moderately ok

Reviewer 4 Report

Comments and Suggestions for Authors

The authors have addressed all comments